# Systemic control of immune cell development by integrated carbon dioxide and hypoxia chemosensation in *Drosophila*

Bumsik Cho[1], Carrie M. Spratford[2], Sunggyu Yoon[1], Nuri Cha[1], Utpal Banerjee[2,3,4] & Jiwon Shim [1,5,6]

*Drosophila* hemocytes are akin to mammalian myeloid blood cells that function in stress and innate immune-related responses. A multi-potent progenitor population responds to local signals and to systemic stress by expanding the number of functional blood cells. Here we show mechanisms that demonstrate an integration of environmental carbon dioxide ($CO_2$) and oxygen ($O_2$) inputs that initiate a cascade of signaling events, involving multiple organs, as a stress response when the levels of these two important respiratory gases fall below a threshold. The $CO_2$ and hypoxia-sensing neurons interact at the synaptic level in the brain sending a systemic signal via the fat body to modulate differentiation of a specific class of immune cells. Our findings establish a link between environmental gas sensation and myeloid cell development in *Drosophila*. A similar relationship exists in humans, but the underlying mechanisms remain to be established.

[1] Department of Life Science, College of Natural Science, Hanyang University, Seoul 04763, Republic of Korea. [2] Department of Molecular, Cell and Developmental Biology, University of California Los Angeles, Los Angeles, CA 90095, USA. [3] Molecular Biology Institute, University of California Los Angeles, Los Angeles, CA 90095, USA. [4] Eli and Edythe Broad Center of Regenerative Medicine and Stem Cell Research, University of California Los Angeles, Los Angeles, CA 90095, USA. [5] Research Institute for Natural Science, Hanyang University, Seoul 04763, Republic of Korea. [6] Research Institute for Convergence of Basic Sciences, Hanyang University, Seoul 04763, Republic of Korea. Correspondence and requests for materials should be addressed to U.B. (email: banerjee@mbi.ucla.edu) or to J.S. (email: jshim@hanyang.ac.kr)

Carbon dioxide ($CO_2$) is the first identified gaseous molecule that evokes innate avoidance behavior in *Drosophila*[1], and is a critical sensory and respiratory cue that alters a variety of animal behaviors[2]. $CO_2$ is detected by a heterodimeric receptor encoded by Gr21a and Gr63a that is expressed in the terminal organ of the larval head or in the antennal olfactory receptor neurons called ab1C in adult flies[3,4]. Even though $CO_2$ was initially identified as a stress molecule, fruits, yeast, and animals emit $CO_2$ as a respiration by-product that lead to complex combinatorial responses to odorants[5].

*Drosophila* hemocytes are akin to mammalian myeloid cells and are sentinels for stress and innate immune-related responses[6,7]. *Drosophila* hemocytes arise from multi-potent blood progenitors and are comprised of three representative classes of myeloid-like cells: plasmatocytes, crystal cells (CCs), and lamellocytes[8]. The majority of mature hemocytes are macrophage-like plasmatocytes while a small fraction becomes CCs known to function in wound healing and innate immune responses[7,9]. Lamellocytes are seldom found in conventional culture conditions and are evident only upon immune challenge[10].

The maintenance of hematopoietic stem- and progenitor populations and their interactions with the niche has been extensively studied in both humans and in model systems[6,8,11,12]. However, the importance of extrinsic cues that originate outside the stem- or progenitor compartment has not been carefully characterized and requires extensive future studies. Complex systemic responses often involve multiple organs and a combination of developmental and stress-related signals[13]. With the use of modern genetic techniques, the *Drosophila* hematopoietic system allows us to delineate mechanistic insights into intricate responses of the myeloid progenitor population to multiple systemic signals[14–16]. However, how sensory neurons that detect the level of ambient gases communicate with the myeloid blood system has not been elucidated although functional analogies have been identified in mammals.

In this study, we identify a genetic link between the respiratory gas-chemosensation and myeloid blood development in *Drosophila*. $CO_2$-sensing and hypoxia-sensing neurons interact at the synaptic level. Low $CO_2$ or $O_2$ triggers the stabilization of Hypoxia inducible factor-α in a small set of neurons in the ventral nerve cord (VNC), promoting transcription of the cytokine *unpaired3* in the brain. This secreted cytokine activates the JAK/STAT pathway in fat bodies (considered similar to the liver), resulting in the expression and secretion of an insulin-like protein, Dilp6. This secreted protein activates the insulin receptor in the hematopoietic organ and this leads to increased levels of the protein Serrate, a ligand for Notch. Increased Notch signaling raises the number of a specific class of immune cells. Notably, this phenotype is recapitulated by modulating atmospheric $CO_2$ or $O_2$, emphasizing that gas perception is directly associated with differentiation of the hematopoietic system in *Drosophila*.

## Results

**Respiratory chemosensation and CC differentiation.** $CO_2$ activates a transmembrane, heterodimeric gustatory receptor complex, called Gr21a/Gr63a[3,4]. It is specifically expressed in the terminal organ of the larval head (Supplementary Fig. 1a). Other tissues, including the hematopoietic organ called the lymph gland (Fig. 1a), do not express this receptor (Supplementary Fig. 1b). The $CO_2$-sensing neuron ($CO_2$SN for simplicity) sends its projection to the subesophageal ganglion (SEG), which in turn connects, through largely unmapped circuits, to the central brain and VNC[3,4,17]. Receptors capable of responding to oxygen levels (or monoxide gases and free radicals) are more widely expressed and belong to the intracellular soluble guanylyl cyclase class of

proteins[18,19]. The receptor Gyc89da is activated by low molecular oxygen ($O_2$) and the multiple neurons expressing it specifically sense hypoxia (HypSNs for simplicity). HypSNs are inhibited in normoxia and hyperoxia[20,21].

We modified the activities of the $CO_2$SN or the HypSNs using a variety of genetic and environmental manipulations (Supplementary Fig. 1c). As a hematopoietic readout, we count CCs, which function in wound healing, clotting, innate immunity, and hypoxic stress response[9,22,23]. Compared with wild-type larvae raised under conventional environmental conditions, we find between 2 and 4-fold increase in the number of CCs upon reduced $CO_2$SN activity (Fig. 1b-l and Supplementary Fig. 1d-l). This phenotype is specific to the CCs and does not alter the number of other cell types or the overall size of the lymph gland (Supplementary Fig. 1m-o). Also, the numbers of sessile and circulating CCs within the larvae are not affected (Supplementary Fig. 1p-s). Loss of HypSN activity has no effect on CC number (Fig. 2a, b, d) while elevated activity of HypSNs causes 2-fold increase in CCs (Fig. 2c, d) when specifically activated in neurons (Fig. 2e–g and Supplementary Fig. 1t). This phenotype is recapitulated upon inhibition of neuronal Gyc89da (Supplementary Fig. 1u). Thus, low $CO_2$SN activity (low $CO_2$ availability) or high HypSNs activity (low $O_2$ availability) favors extra CC formation.

The increased CC phenotype due to low $CO_2$SN activity is fully suppressed to wild-type numbers by concurrent low HypSN activity (Fig. 2h–k), raising the possibility of a coupled response. An RNAi-based mini-screen for enzymes that synthesize neurotransmitters revealed that knockdown of *Gad1* (encoding the GABA synthesis enzyme, glutamate decarboxylase) in the $CO_2$SN gives increased CC numbers (Fig. 2l and Supplementary Fig. 1v). Remarkably, knockdown of the GABA$_B$ receptors, R1/R2 in the HypSNs similarly raises CC numbers (Fig. 2m and Supplementary Fig. 1w, x). This prompted us to investigate a possible direct interaction between these neurons at inhibitory synapses.

We expressed the dendritic marker, DenMark[24], in the $CO_2$SN and simultaneously marked HypSNs with GFP. Separate neurons with nuclei residing within the terminal ganglion express these two receptors. Although both send anterior projections to the terminal organ, at this location, they appear to be non-overlapping (Supplementary Fig. 1y). In contrast, the posterior projection from the single $CO_2$SN approaches the SEG, where it comes into extremely close association with a projection from HypSNs (Fig. 2n).

We next utilized the GRASP technique[25,26], in which a positive fluorescence signal indicates molecular level proximity between the two neurons. Using membrane-GRASP, we detect such close association between the $CO_2$SN and the HypSNs at the level of the SEG (Supplementary Fig. 1z, aa). For even finer analysis, we used Synaptobrevin::GRASP that would only highlight points of active synaptic contacts (Full genotype: *Gyc89da-gal4; Gr63a-LexA, UAS-CD4-spGFP$_{11}$, LexAop-nSyb-spGFP$_{1-10}$*). Punctate signals are readily evident in the anterior SEG indicating synapse formation between these two classes of neurons (Fig. 2o, p). Taken together, the genetic data on the involvement of GABAergic neurons, the DenMark data on proximity of labeled branches and the GRASP analyses, we conclude that the $CO_2$SN forms inhibitory synapses with HypSN branches at the level of the SEG. These data do not preclude additional parallel interactions elsewhere within the neuronal circuitry of the central brain.

**Attenuation of $CO_2$SN stabilizes Hifα in HypSNs in the VNC.** Hypoxic conditions allow stabilization of Hypoxia inducible factor-α (Hifα, called Sima in *Drosophila*) and favor CC

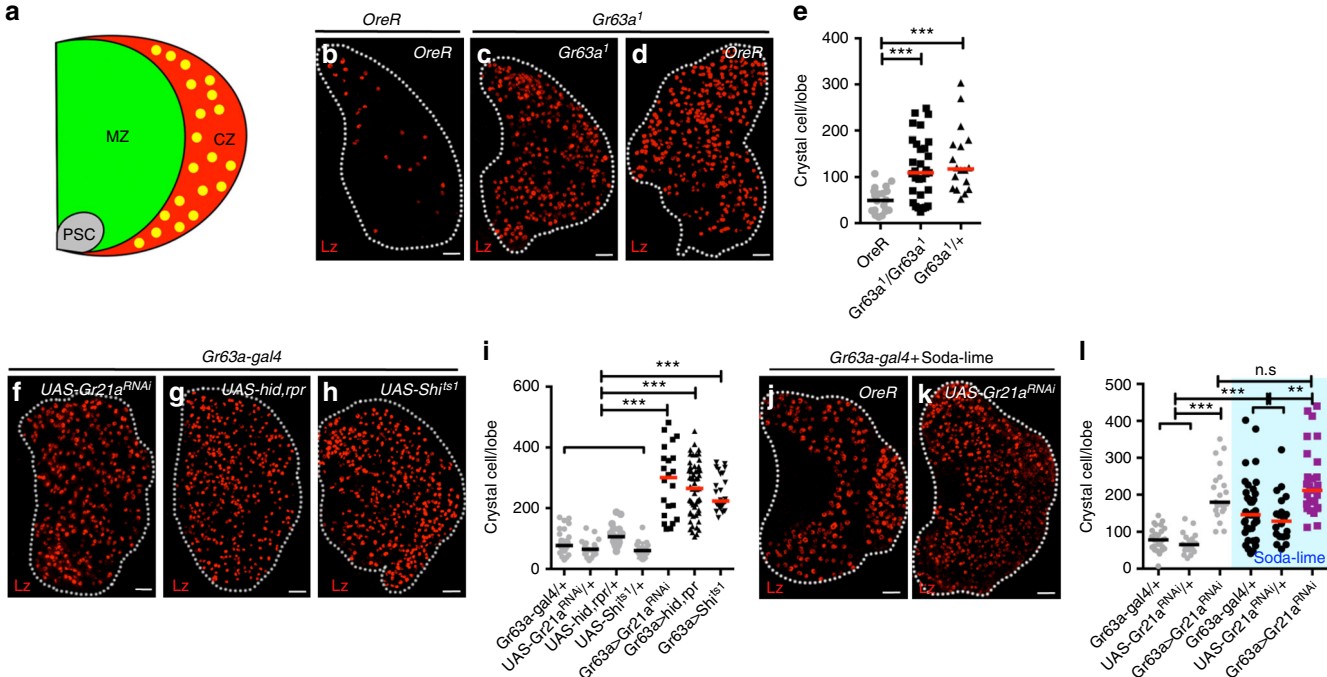

**Fig. 1** $CO_2$ chemosensation controls the differentiation of crystal cells. Graphs indicate the number of crystal cells (CC) per lymph gland lobe. Bars in graphs: the median. n.s: not significant ($p > 0.01$). *$p < 0.01$; **$p < 0.001$; ***$p < 0.0001$. Scale Bar: 20μm. White dotted lines: lymph glands. Statistical results and genotypes are indicated in Supplementary Table 3. **a** A schematic representation of the primary lobe of the hematopoietic organ, the larval lymph gland. Undifferentiated blood progenitors reside in the inner core, termed the medullary zone (MZ), and give rise to mature blood cell lineages including crystal cells (CC, marked in yellow) that occupy a region termed the cortical zone (CZ). The posterior signaling center (PSC) secretes multiple factors to maintain the progenitors. Only the features relevant to this study are shown. **b–l** Loss of $CO_2SN$ activity causes increased CC number. Wild-type third-instar larvae express fewer than 100-CCs per primary lymph gland lobe (CCs are marked in red, Lz) (**b**) Both homozygous ($Gr63a^1/ Gr63a^1$) (**c**) and heterozygous ($Gr63a^1/+$) mutants (**d**) of $Gr63a$, the gene encoding the chemoreceptor for $CO_2$ sensing, exhibit increased numbers of CCs. Quantitation of CC numbers shown in (**e**). $Gr21a$ encodes the second subunit of the $CO_2$ chemoreceptor that functions with $Gr63a$. Knockdown of $Gr21a$ function ($Gr63a$-gal4; $UAS$-$Gr21a^{RNAi}$) (**f**), or genetic ablation of the $CO_2$ receptor neuron by expression of pro-apoptotic genes, $hid$ and $rpr$ ($Gr63a$-gal4; $UAS$-$hid,rpr$) (**g**) or attenuation of synaptic transmission by expression of a temperature-sensitive form of the dynamin-like protein Shibire ($Gr63a$-gal4; $UAS$-$Shi^{ts1}$) (**h**), each causes a significant increase in the number of CCs. This phenotype is not observed in Gal4- or UAS-controls alone. Quantitation is shown in **i**. Absorption of ambient $CO_2$ in larval culture vial with the use of soda-lime, a mixture of bases that eliminates gaseous $CO_2$ (see Methods for detail), mimics the $CO_2SN$ mutant phenotype (**j**). This phenotype is enhanced by a simultaneous knockdown of $Gr21a$ in the $CO_2SN$ ($Gr63a$-gal4; $UAS$-$Gr21a^{RNAi}$) (**k**). Quantitation shown in **l**. The blue shading in panels (**l**) represents soda-lime-induced low environmental $CO_2$ condition

differentiation through non-canonical activation of Notch[27]. A small number of cells in the VNC express very low levels of Sima protein even in wild-type larvae grown under normoxic conditions (Fig. 3a) and this Sima protein expression is upregulated if $CO_2SN$ activity is attenuated (Fig. 3b, c), or if HypSN activity is increased (Fig. 3d–f). Additionally, loss of $Gad1$ in the $CO_2SN$ raises Sima levels in the VNC neurons (Supplementary Fig. 2a-c), highlighting the interaction between $CO_2SN$ and HypSNs in this process. Overexpression of $sima$ in HypSNs is sufficient to induce increased CC differentiation under normal gaseous ligand sensation (Fig. 3g and Supplementary Fig. 2d). Also, knockdown of $sima$ transcript in HypSNs rescues the CC phenotype seen under low $CO_2SN$ activity (Fig. 3h, i and Supplementary Fig. 2e-g). These two results are also seen if $sima$ levels are manipulated specifically in all neurons (Fig. 3j, k and Supplementary Fig. 2g-i). Thus, no non-neuronal participant is essential and Sima increase in HypSNs is both necessary and sufficient for linking the sensory signals to CC formation. The high VNC Sima expression is seen in 14 pairs of neurons of which 10 pairs are HypSNs (Fig. 3l, m). These results establish that HypSNs that are also Sima$^+$ are important for the CC phenotype.

**Upd3 from the brain signals to the fat body**. VNC neurons are known to secrete hormones and cytokines into the hemolymph[28].

We conducted a targeted mini-screen of known hormone and secreted factor-encoding genes (Supplementary Fig. 3a) to determine if any of these is differentially expressed at the mRNA level in the $Gr63a$ mutant brain compared with controls. Three of these genes are upregulated in the mutant at the transcriptional level, of which only one, that encodes the cytokine $unpaired3$ ($upd3$) gives increased CC numbers upon pan-neuronal overexpression (Fig. 4a–c and Supplementary Fig. 3b). Loss of $upd3$ in neurons does not cause any hematopoietic defect (Fig. 4d) but $upd3$-RNAi or $upd3$ mutants suppress the extra CC phenotype of $CO_2SN$ inhibition (Fig. 4e, f and Supplementary Fig. 3c-f). This indicates a role for Upd3 in transmitting the gaseous ligand-generated stress signal to regions outside the brain.

Under conditions of reduced $CO_2SN$ activity, $upd3$ is detected in a large number of cells in the brain that are not obligatorily HypSNs or Sima$^+$ (Fig. 4g, h). Yet, $upd3$ and HypSNs co-localize and synapse onto each other at the posterior region of VNC (Full genotype: $Upd3$-gal4, $UAS$-mCherry; $Gyc89da$-LexA, $UAS$-CD4-spGFP$_{11}$, LexAop-nSyb-spGFP$_{1-10}$) (Fig. 4i–k). Consistently, experimental evidence presented below suggests that the observed increase in $upd3$ is critically dependent on Sima$^+$ HypSNs. Overexpression of $sima$ in HypSNs is sufficient to induce a 4-fold increase in $upd3$ transcription in the brain (Fig. 4l). Also, loss of $sima$ in HypSNs rescues the high $upd3$ transcription in a reduced

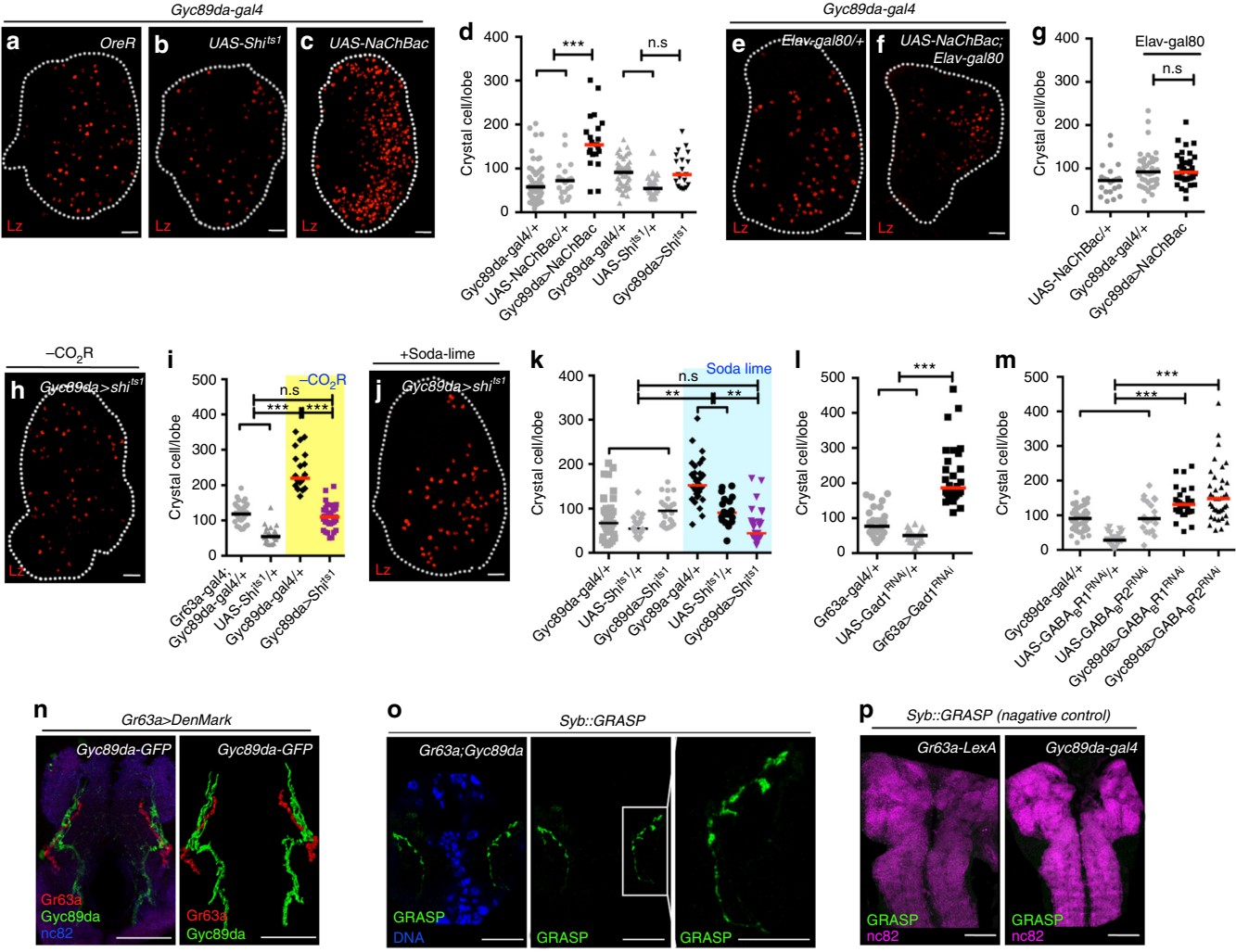

**Fig. 2** $CO_2SN$ forms inhibitory synapses with HypSN. Graphs indicate the number of crystal cells (CC) per lymph gland lobe. Bars in graphs: the median. n. s: not significant ($p > 0.01$). *$p < 0.01$; **$p < 0.001$; ***$p < 0.0001$. Scale Bar: 20 µm, unless otherwise indicated. White dotted lines: lymph glands. $CO_2SN$ inhibited conditions: −$CO_2R$ with shaded yellow in **i**, and Soda-lime generated low $CO_2$: shaded blue in **k** for clarity. **a–g** Constitutive activation of HypSNs induces CC differentiation. Control animals (*Gyc89da-gal4/ +*) have an average of 70-90 CCs (**a**). Inactivation of HypSNs (*Gyc89da-gal4; UAS-Shi$^{ts1}$*) does not alter CC number (**b**). Constitutive activation of HypSNs by expressing the bacterial sodium channel, NaChBac (*Gyc89da-gal4; UAS-NaChBac*) results in increased CC differentiation (**c**). Quantitation is shown in **d**. When the expression of NaChBac is specifically blocked in neurons (*Gyc89da-gal4, Elav-gal80; UAS-NaChBac*) the average CC number is unchanged (**f**) from that in control (**e**). Quantitation shown in **g**. **h–k** HypSNs function downstream of $CO_2SN$. As shown in (Fig. 1h), $CO_2SN$ inhibition increases CC number (*Gr63a-gal4; UAS-Shi$^{ts1}$*). This phenotype reverts to wild type when HypSNs are also inhibited (*Gr63a-gal4; Gyc89da-gal4, UAS-Shi$^{ts1}$*) (**h**). Quantitation shown in **i**. Soda-lime-mediated CC phenotype is also alleviated by inhibition of HypSNs (low atmospheric $CO_2$ + *Gyc89da-gal4; UAS-Shi$^{ts1}$*) (**j**). Quantitation shown in **k**. **l–m** GABA-mediated inhibition of HypSNs by $CO_2SN$. Knockdown of *Gad1* in the $CO_2SN$ (*Gr63a-gal4; UAS-Gad1$^{RNAi}$*) enhances differentiation of CCs (**l**). Similarly, loss of either *GABA$_B$R1* or *GABA$_B$R2* in HypSNs (*Gyc89da-gal4; UAS-GABA$_B$R1$^{RNAi}$* or *Gyc89da-gal4; UAS-GABA$_B$R2$^{RNAi}$*) also leads to increased CC differentiation (**m**). **n–o** Physical proximity and overlap of $CO_2SN$ and HypSNs in the SEG. Projections of the $CO_2SN$ adjoining HypSNs at the level of the SEG shows significant co-localization: $CO_2SN$ in red and HypSN in green (*Gr63a-gal4; UAS-DenMark; Gyc89da-GFP*). Rendered image of the co-localization data shown in right panel (**n**). Syb::GRASP expression of the $CO_2SN$ and HypSNs in the SEG (**o**). Syb::GRASP signal resulting from points of contact is shown in green (Magnified images in the following panels) (**o**). **p** Negative control experiments for the Syb::GRASP data shown in **o**. *Gr63a-LexA* alone (*Gr63a-LexA; UAS-CD4-spGFP$_{11}$, LexAop-nSyb-spGFP$_{1-10}$*) or *Gyc89da-gal4* alone (*Gyc89da-gal4; UAS-CD4-spGFP$_{11}$, LexAop-nSyb-spGFP$_{1-10}$*) does not give rise to a GRASP signal. Scale bar: 50 µm

$CO_2SN$ activity background (Fig. 4m). Simultaneous loss of $CO_2SN$ and HypSN activities gives wild-type levels of *upd3* transcription (Fig. 4m).

Socs36e is a direct downstream transcriptional target of the JAK/STAT signaling pathway initiated by Upd3 upon binding its receptor, Domeless (Dome)[29]. qPCR analysis of dissected tissues from larvae lacking $CO_2SN$ activity shows upregulation of *Socs36e* specifically in the fat body (considered similar to the liver) (Fig. 4n), and importantly, not in the lymph gland (Supplementary Fig. 3g).

Fat body *Socs36e* expression is suppressed when either: *upd3* is down-regulated in the brain, or when *sima* expression is decreased within HypSNs, or upon simultaneous inhibition of $CO_2SN$ and HypSNs (Fig. 4o, p). Additionally, overexpression of either *sima* or *upd3* in the brain is sufficient to induce *Socs36e* in the fat body (Supplementary Fig. 3h, i). Finally, *dome-RNAi* autonomously suppresses *Socs36e* in the fat body and non-autonomously affects CC number in $CO_2SN$ activity-depleted larvae (Fig. 4q–s and Supplementary Fig. 3j, k).

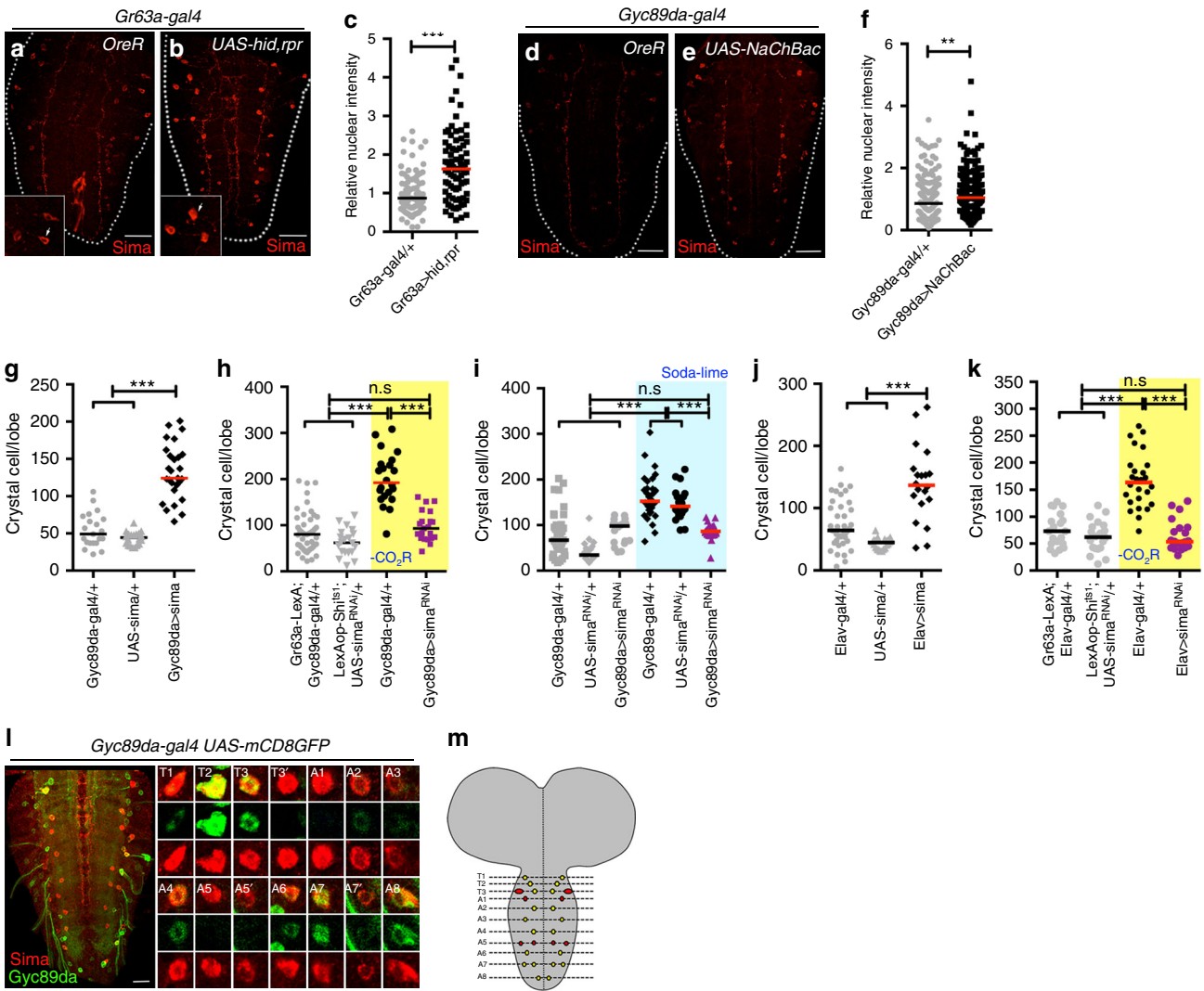

**Fig. 3** Sima (Hifα) stabilization in HypSNs affects crystal cell number. **c** and **f** indicate the relative intensities of nuclear Sima, and Graphs in **g–k** indicate the number of CCs in a single lymph gland lobe. n.s: not significant ($p > 0.01$). *$p < 0.01$; **$p < 0.001$; ***$p < 0.0001$. Scale Bar: 50μm. Bars in graphs: the median. $CO_2$SN inhibited conditions: –$CO_2$R with shaded yellow in **h, k**, and Soda-lime generated low $CO_2$: shaded blue in **i** for clarity. **a–c** Inhibition of the $CO_2$SN elevates Sima expression in the VNC. Control brains (*Gr63a-gal4/ +*) exhibit very low Sima expression in the VNC (Sima is in red) (**a**). Genetic ablation of $CO_2$SN (*Gr63a-gal4; UAS-hid,rpr*) leads to 2-fold increase in accumulation of nuclear Sima in specific VNC cells (white arrows in inset) (**b**). Quantitation of data shown in **c**. **d–i** Activation of HypSNs elevates Sima expression in the VNC. Control brains (*Gyc89da-gal4/ +*) show very low Sima expression (**d**). Constitutive activation of HypSNs significantly increases accumulation of nuclear Sima in specific VNC cells (*Gyc89da-gal4; UAS-NaChBac*) (**e**). Quantitation of relative nuclear Sima intensity shown in **f**. CC differentiation is significantly enhanced when *sima* is overexpressed in HypSN cells (*Gyc89da-gal4; UAS-sima*) (**g**). *sima*RNAi expressed in HypSNs suppresses the extra CC differentiation phenotype caused by loss of $CO_2$SN function (*Gr63a-LexA, LexAop-Shi*ts1*; Gyc89da-gal4, UAS-sima*RNAi) (**h**). This rescue is also seen upon loss of *sima* in HypSNs when animals are grown in low $CO_2$ (soda-lime) condition (**i**). **j-k** *sima* expression in neurons is linked to CC differentiation. Larvae expressing *sima* in the brain (*Elav-gal4; UAS-sima*) facilitates CC differentiation (**j**). Also, increased CC differentiation caused by $CO_2$SN inhibition is rescued by concurrent inhibition of *sima* in the brain (*Gr63a-LexA, LexAop-Shi*ts1*; Elav-gal4, UAS-sima*RNAi) (**k**). **l**, **m** Sima co-localizes with a subset of HypSNs. Within the VNC, neurons express Sima in total 14 pairs per hemineuromere from T1 to A8. Amongst these, 10 pairs co-localize with Gyc89da in the T1-T3, A2-A4 and A6-A8 (*Gyc89da-gal4; UAS-mCD8GFP*). A magnified view of Sima and Gyc89da-expressing neurons in the thoracic and abdominal ganglia (Sima, red; Gyc89da, green) (**l**). In the reconstructed image of VNC, Sima+ HypSNs are shown in yellow and Sima+ cells that are not HypSNs are marked in red (**m**)

## Dilp6 induces Serrate via InR in blood progenitors.

We screened for increased expression of RNAs encoding neurosecretory peptides upon loss of $CO_2$SN activity. Only one positive candidate, the *Drosophila Insulin-like peptide-6* (*dilp6*), is normally low in the fat body during larval stages[30] and is specifically upregulated in this organ when the $CO_2$SN is mutated (Fig. 5a–d and Supplementary Fig. 4a-d). While *dilp6* is additionally expressed in glia[31], we found that overexpression in the fat body, but not in glia, has an effect on CC number (Fig. 5e and Supplementary Fig. 4e-g). Similarly, the CC phenotype due to

reduced $CO_2$SN activity is efficiently rescued when *dilp6* is specifically blocked in the fat body or in the *dilp6*[41] mutant, but not when *dilp6-RNAi* is expressed in glial cells (Fig. 5f, g and Supplementary Fig. 4h-k).

Overexpression of *sima* or *upd3* in the brain raises fat body levels of *dilp6* (Fig. 5h, i). Increased *dilp6* in a $CO_2$SN activity-depleted background is suppressed: upon concurrent inhibition of HypSNs, with attenuation of brain *sima* or *upd3*, or when *dome-RNAi* is expressed in the fat body (Fig. 5j-l). A different insulin related peptide, Dilp2, functions in hematopoiesis, but not

directly in CC formation[14,32] and no previously known role in hematopoiesis was identified for Dilp6. Both Dilps function by binding to the insulin receptor (InR), which is known to promote differentiation of CCs[16,32]. In $CO_2SN$ activity-depleted larvae, we

detect high pAKT and p4EBP, direct phosphorylation targets of the InR pathway in the lymph gland (Fig. 5m–p and Supplementary Fig. 4l-q). This is also seen when *dilp6* is overexpressed in the fat body (Supplementary Fig. 4r, s). We

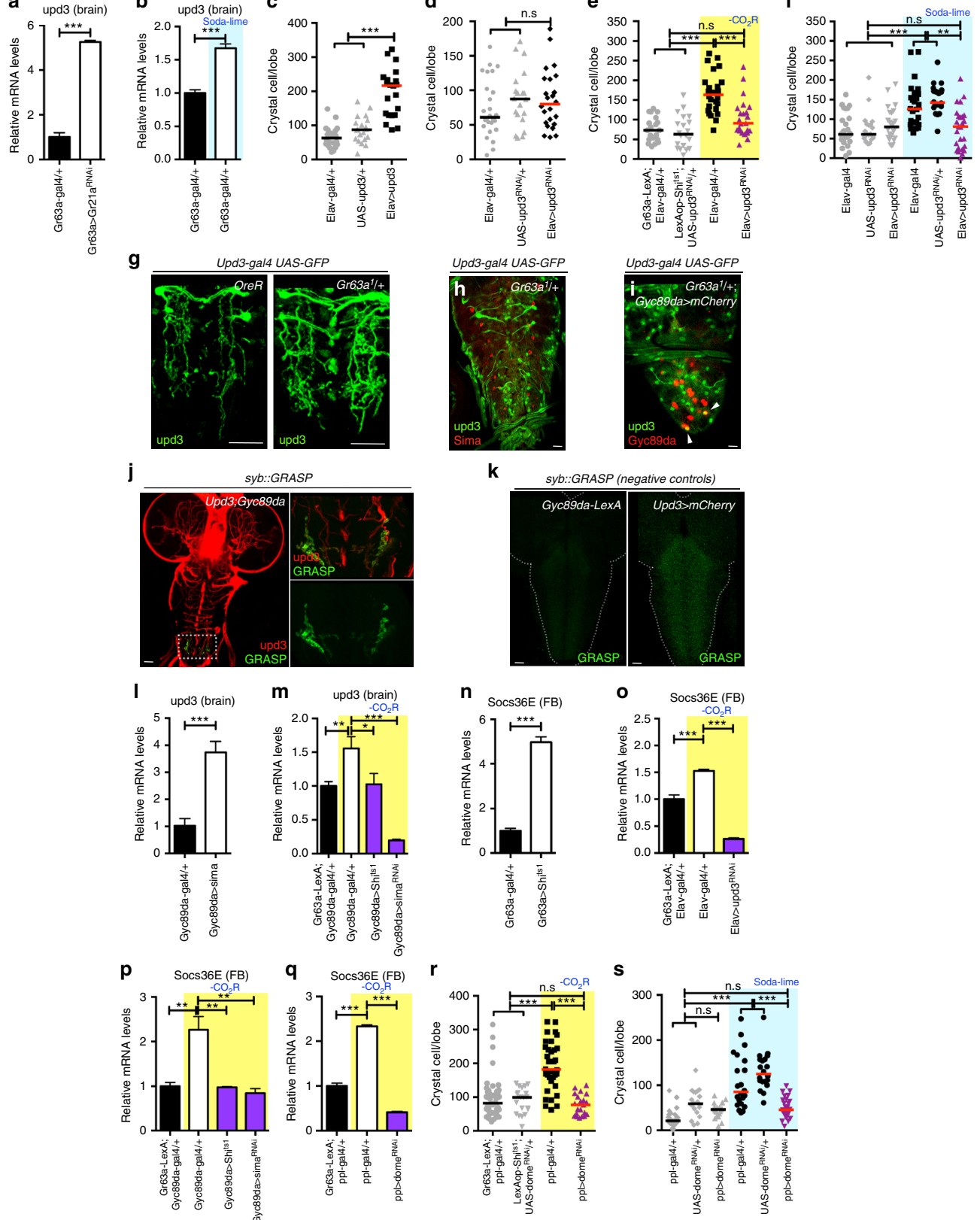

hypothesize that altered gaseous signaling leads to systemic secretion of Dilp6 from the fat body that activates InR pathway[30,31,33].

A hallmark of CC fate specification is the interaction between the ligand Serrate, expressed in internal signaling centers within the lymph gland, with its receptor Notch, present in neighboring cells[27,34,35]. Wild-type lymph glands from mid-second instar larvae exhibit low Serrate protein expression at the edge where differentiation is initiated (Fig. 5q). Reduced $CO_2SN$ activity has a pronounced effect on membrane Serrate expression as this protein is detected in more cells at a much higher level when compared with wild type (Fig. 5r and Supplementary Fig. 4t-v). Increased Serrate is also seen upon *sima* or *upd3* overexpression in the brain, or when *dilp6* is driven in the fat body (Fig. 5s–u). We also detect an increase in number of cells in which Notch is active, but there is no change in the activation level per cell (Fig. 5v–x). We conclude that the cascade leading up to InR increases Serrate expression, which in turn activates the Notch pathway in an increased number of cells causing them to take on CC fates.

### CC differentiation controlled by altered environmental gases.

A model summarizing the results presented in this study is shown in Fig. 6a. Environmental $CO_2$ activates its receptor Gr21a/Gr63a expressed on the $CO_2SN$, while environmental levels of $O_2$ repress HypSNs. This occurs at the level of sensory neurons, and the information is relayed to higher brain centers. Genetic data support communication between these sensory neurons in the suboesophageal ganglion (SEG) and accumulation of Sima in a different set of HypSNs within a small number of VNC neurons. The neuronal circuits between the SEG, VNC and higher brain centers have not been mapped yet. However, it is clear that Sima enhances *upd3* expression in the brain and that secretion of this cytokine activates the JAK/STAT pathway in the fat body and this results in Dilp6 expression and secretion into the hemolymph. The resulting Dilp6/InR signal within the lymph gland causes an increase in the level and number of cells that express Serrate. As is well known from the published literature, the Serrate–Notch interaction is critical in determining CC fate and number within the lymph gland[19,26]. The interorgan communication system identified in this study allows the monitoring of ambient gas levels in the environment and through their integration, allows any chronic imbalance of respiratory gases for the purpose of

proper stress response and the maintenance of immune homeostasis.

A unique feature of this model is the integration of $CO_2$ and $O_2$ sensation to achieve a common blood phenotype. For the results to be physiologically relevant, alteration of environmental gaseous ligands should phenocopy the effects of genetic manipulations. The normal atmosphere contains approximately 0.04% of $CO_2$ with additional $CO_2$ released from fermenting food sources. Environmental $CO_2$ is scavenged in larval culture vials with the use of a mixture of bases, "soda-lime" (see Supplementary Fig. 1c and Methods). The soda-lime method scavenges atmospheric $CO_2$ to very low levels without altering development, while hypoxia chambers can create controlled hypoxia, hyperoxia, or hypercapnia environments. Most importantly, exposure of wild-type larvae to either soda-lime or to hypoxia increases CC numbers (Fig. 6b, c and Supplementary Fig. 5a, b compare with the genetic manipulations in Fig. 1i). Lowering $CO_2SN$ activity does not further enhance the hypoxia phenotype (Fig. 6c and Supplementary Fig. 5c). On its own, hyperoxia decreases CC numbers compared to wild type and also suppresses the low $CO_2SN$-activity phenotype (Fig. 6d and Supplementary Fig. 5d, e compare Fig. 2d, i), whereas hypercapnia does not alter the CC formation (Supplementary Fig. 5f). Finally, soda-lime and hypoxia both: increase nuclear Sima in neurons (Fig. 6e-h compare Fig. 3a-f), and increase *upd3* in the brain (Fig. 6i–k compare Fig. 4a, b), and *Socs36e* (Fig. 6l–n compare Fig. 4n, p) and *dilp6* in the fat body (Fig. 6o–q compare Fig. 5c, j). These results demonstrate that the CC phenotypes can be triggered by imbalances in respiratory gases in wild-type animals, establishing a physiological relevance for our observations.

## Discussion

A wild-type number of CCs is generated through local developmental signals independent of sensory input[27,34,35]. The multi-organ and multi-pathway cascade described here represents a stress signal activated upon alteration in respiratory gases over the developmental time period. Such sustained variations in gaseous components are likely to be experienced fairly often during larval development. During the time period over which hematopoiesis is at its peak within the lymph gland, larvae experience hypoxic conditions buried into the food that they forage through. Decomposition and yeast (a primary food source) cause variations in $CO_2$ levels. Toward the end of the

---

**Fig. 4** Upd3 secreted from the brain triggers a systemic signal. **c–f**, **r–s** indicate CC numbers in a single lymph gland. n.s: not significant ($p > 0.01$). *$p < 0.01$; **$p < 0.001$; ***$p < 0.0001$. Error bars in **a–b**, **l–q**: standard deviation. Bars in **c–f**, **r–s**: the median. Scale Bar: 50μm. $CO_2SN$ inhibited conditions: $-CO_2R$ with shaded yellow in **e, m, o-r**. Soda-lime generated low $CO_2$ conditions: shaded blue in **b, f, s**. **a-b** *upd3* upregulation in the brain is linked to CC differentiation. Loss of Gr21a (*Gr63a-gal4; UAS-Gr21a$^{RNAi}$*) in the $CO_2SN$ (**a**) or scavenging of $CO_2$ (**b**) results in increased *upd3* mRNA in the brain. **c-m**. Upd3 functions downstream of Sima. *upd3* in the brain causes increased CC formation (*Elav-gal4; UAS-upd3*) (**c**). Loss of neuronal *upd3* (*Elav-gal4; UAS-upd3$^{RNAi}$*) alone does not alter CC number (**d**). Silencing neuronal *upd3* suppresses the CC phenotype caused by inhibition of $CO_2SN$ (*Gr63a-LexA, LexAop-Shi$^{ts1}$; Elav-gal4, UAS-upd3$^{RNAi}$*) (**e**) or by soda-lime treatment (*Elav-gal4; UAS-upd3$^{RNAi}$*) (**f**). Compared with control, *upd3* is elevated in the posterior VNC of *Gr63a$^1$/+* mutants (*Upd3-gal4, UAS-GFP; Gr63a$^1$/+*) (**g**). No overlap is observed between Sima$^+$ and *upd3$^+$* neurons (**h**). HypSN seen in the posterior VNC are occasionally *upd3*-positive (white arrow heads), but the majority of *upd3$^+$* cells are not HypSNs (*Upd3-gal4, UAS-GFP; Gyc89da-LexA; LexAop-mCherry*) (**i**). Syb::GRASP expression of the HypSNs and Upd3$^+$ neurons in the VNC (Magnified images in the following panels) (Upd3, red; GRASP, green) (**j**). *Gyc89da-LexA* alone (*Gyc89da-LexA; UAS-CD4-spGFP$_{11}$, LexAop-nSyb-spGFP$_{1-10}$*) or *Upd3-gal4* alone (*Upd3-gal4 UAS-mCherry; UAS-CD4-spGFP$_{11}$, LexAop-nSyb-spGFP$_{1-10}$*) does not give rise to a GRASP signal (**k**). Overexpression of *sima* in HypSNs increases neuronal *upd3* mRNA (*Gyc89da-gal4; UAS-sima*) (**l**). *sima* RNAi or inhibition of HypSNs suppresses the elevated *upd3* levels (*Gr63a-LexA, LexAop-Shi$^{ts1}$; Gyc89da-gal4, UAS-sima$^{RNAi}$ or Gr63a-LexA, LexAop-Shi$^{ts1}$; Gyc89da-gal4, UAS-Shi$^{ts1}$*) (**m**). **n-s** Brain-secreted Upd3 functions in the fat body. *Socs36e* is induced in the fat body upon loss of $CO_2SN$ (*Gr63a-LexA; LexAop-Shi$^{ts1}$*) (**n**). This expression is suppressed upon: loss of neuronal *upd3* (*Gr63a-LexA, LexAop-Shi$^{ts1}$; Elav-gal4, UAS-upd3$^{RNAi}$*) (**o**), by simultaneous inhibition of HypSNs (*Gr63a-LexA, LexAop-Shi$^{ts1}$; Gyc89da-gal4, UAS-Shi$^{ts1}$*) (**p**), by *sima* RNAi in the HypSNs (*Gr63a-LexA, LexAop-Shi$^{ts1}$; Gyc89da-gal4, UAS-sima$^{RNAi}$*) (**p**), or upon loss of *dome* in the fat body (*Gr63a-LexA, LexAop-Shi$^{ts1}$; ppl-gal4, UAS-dome$^{RNAi}$*) (**q**). *dome* RNAi in the fat body reverts the CC numbers in the $CO_2SN$ mutant (*Gr63a-LexA, LexAop-Shi$^{ts1}$; ppl-gal4, UAS-dome$^{RNAi}$*) (**r**) or in soda-lime conditions (*ppl-gal4; UAS-dome$^{RNAi}$*) (**s**) to wild-type

hematopoietic developmental period, larvae spend extended periods of time in a very different environment awaiting pupariation. Finally, molting is associated with the shedding of the cuticular intima and degeneration of tracheoles that lack these inner linings[36]. As the tracheal tube is filled with fluid and devoid of gases at this stage, we speculate that this process is also likely to alter oxygen tension in the hemolymph. In past studies, CCs have been associated with hypoxia and hypoxic stress as well as innate immune response[23,27,37]. How increased numbers of these cells will mitigate the effects of gaseous imbalance will require detailed analysis in the future. However, in order to determine whether

the presence of CCs provides a benefit to the whole animal, we generated flies in which the final step in the cascade, Serrate, is eliminated in cells from which CCs are derived during larval development. We then tested the emerging adults for sensitivity to hypoxia and found that these flies fully paralyze in a hypoxia chamber much more readily than genetically matched control flies (Supplementary Fig. 5g, h; the $p$-value is <0.0001). Thus, in addition to their other functions, CCs provide general protection against hypoxia to the animal. For myeloid progenitors, which are sentinels for stress and infection, we consistently find that stress signals feed into developmental pathways, in this case

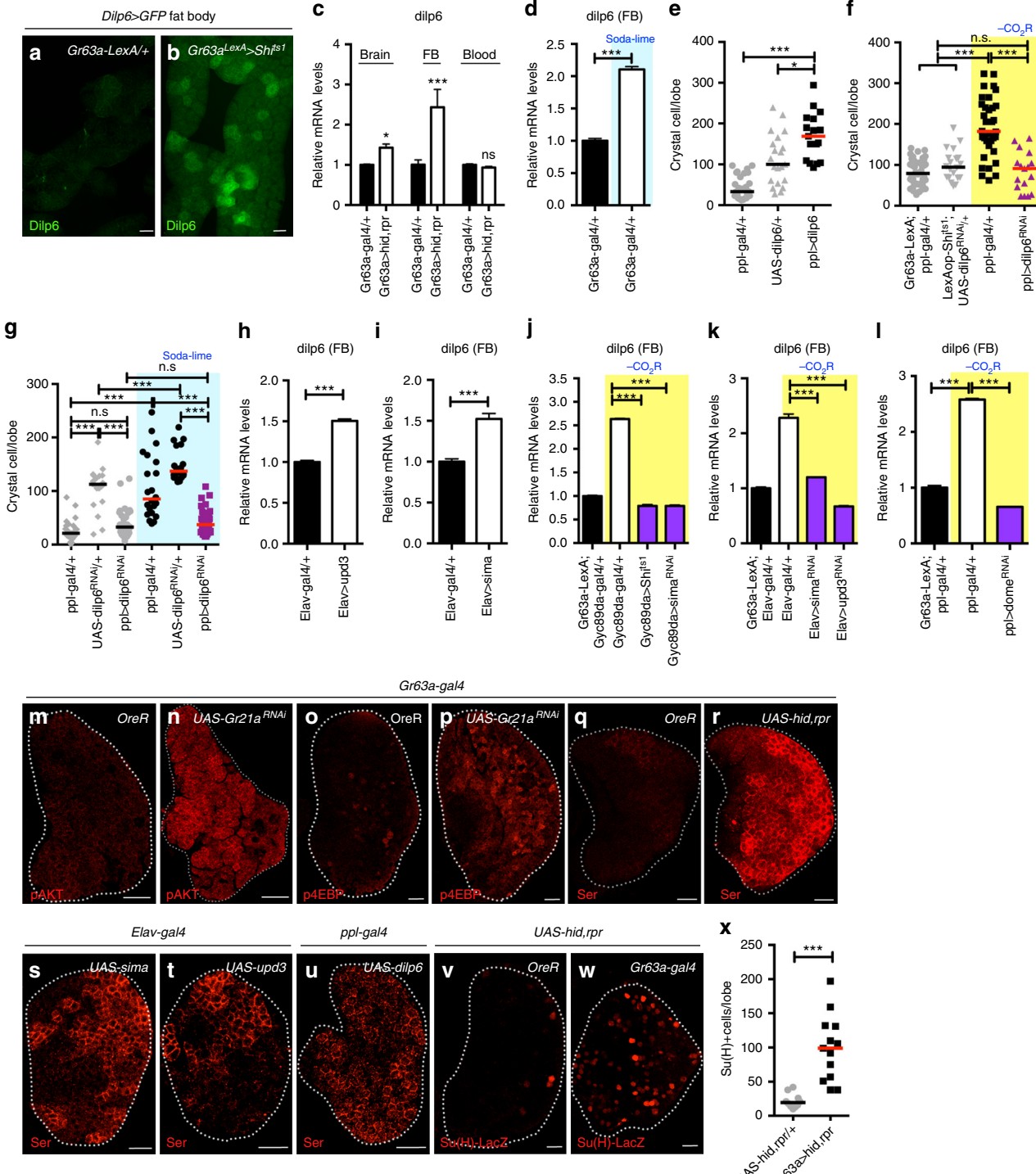

Serrate–Notch signaling, to enhance the homeostatic response to a level more appropriate for rapid immune and stress response[9,38].

To determine if loss of gaseous sensation is linked to innate immunity, we tested levels of antimicrobial peptides in animals lacking CO2SN. Indeed, loss of $CO_2$ sensation is associated with a four to six-fold increase in the transcription of *Drosomycin* and a four-fold increase for *Drosocin* (Supplementary Fig. 5i). However, the significance of this response by the blood cells in the absence of any microbial infection is not clear. We attribute this global and preemptive augmentation of the innate immune system to the increased concentrations of cytokines such as Upd3 and Dilp6 that result from long-term loss of gaseous sensation. Additional physiological effects such as altered lifespan in *Drosophila*[39] and innate immune response in *C. elegans*[40,41] have been associated with $CO_2$ and $O_2$ sensation. Together, these studies allow us to infer that $O_2$ and $CO_2$ chemosensation has a conserved role in animal physiology and immunity.

Although the mechanistic details are not yet deciphered, it seems clear that this conservation extends to mammalian species including humans. Several studies suggest a crosstalk between $CO_2$ and $O_2$ in mammals[42] and establish an influence of gaseous sensation on the hematopoietic system[43,44]. The ventral surface of the mammalian medulla oblongata senses $CO_2$[45] and responds to $O_2$ sensing by the carotid chemoreceptor neurons[46,47]. Chemosensation and immunity are closely linked and each is evolutionarily conserved at a mechanistic level. Whether a multi-organ cascade involving multiple cytokines similar to that described in this study links gaseous signaling to myeloid cell function and development in humans will be attractive to investigate.

## Methods

**Drosophila stocks and genetics.** The following *Drosophila* stocks were used in this study: *Gr63a-gal4* (BL9942), *Gr21a-gal4* (BL24147), *Gyc89da-gal4* and *Gyc89da-GFP* (D. Morton), *Elav-gal4* (BL8765), *ppl-gal4* (BL58768), *HHLT-gal4* (C. Evans), *Upd3-gal4* (H. Agaisse), *Repo-gal4* (BL7415), *Dilp6-gal4* (A. Brand), *HmlΔ-gal4* (S. Sinenko), *Hml-dsRed; Dome-Meso-GFP* (U. Banerjee), *Gr21a RNAi* (BL31281 and VDRC104122), *dilp6 RNAi* (BL33684 and VDRC102465), *upd3 RNAi* (VDRC106869), *sima RNAi* (VDRC106187), *dome RNAi* (VDRC19717), *Gad1 RNAi* (BL51794), *GBR1 RNAi* (VDRC101440), *GBR2 RNAi* (BL50608), *Serrate RNAi* (VDRC27172), *UAS-sima* (BL9582), *UAS-dilp6* (E. Hafen), *UAS-upd3* (B. Lemaitre), *UAS-hid, rpr* (Nambu JR), *UAS-syb::GRASP* (BL64315), *UAS-CD4::GRASP* (BL58755), *UAS-GTrace* (C. Evans), *UAS-NaChBac* (BL9469), *UAS-mCD8GFP* (BL5137), *UAS-DenMark* (BL33063), *Elav-gal80* (Y.N. Jan), *LexOp-Shi^{ts1}* (G. Rubin), *13XLexAop2-6XmCherry-HA* (BL52271), *12xSu(H)-LacZ* (S. Artavanis-Tsakonas), *Gr63a^1* (BL9941), *Df[Gr21a]* (DGRC150003), *dilp6^{41}* (BL30885), *upd2Δupd3Δ* (BL55729), *upd3Δ* (BL55728).

Generation of *Gr63a-LexA*, *Gyc89da-LexA* and *Lz-LexA* flies: *Gr63a* enhancer[4], *Gyc89da* enhancer[20] or *Lz* enhancer (Forward primer sequence:GGGATTAGGC-AGTGTTCCC, Reverse primer sequence:GTACCAATCGCTCCATCCAC) was amplified from fly genomic DNA and ligated into the TOPO-TA vector (Invitrogen) for Gateway cloning. Each entry vector was ligated into the *pBPnlsLexA::p65Uw* (Addgene 26230) destination vector using the LR ligase (Invitrogen). Transgenic flies were generated by BestGene Inc.

All fly stocks were maintained at 18 °C. Unless indicated, crossed flies were maintained at 29 °C with dextrose-cornmeal based conventional food for maximum Gal4-UAS/LexA-LexAoP expression. Experiments with soda-lime/hypoxia/hyperoxia/hypercapnia and synchronization[15] of larvae were performed at 25 °C. *Gyc89da-gal4* crossed with *UAS-sima* flies were maintained at 18 °C until reaching the mid-second instar (approximately 5 days) and shifted to 25 °C. *Elav-gal4* crossed with *UAS-sima* or *UAS-upd3*, or *Gyc89da-gal4* crossed with *UAS-NaChBac* was maintained at 25 °C. These above four genotypes show a drowning or lethal phenotype at 29 °C. *Gr63a^1* mutants were back-crossed more than 50 generations. *Gyc89da-gal4* or *Dilp6-gal4* was recombined with *UAS-mCD8GFP*; *Gyc89da-LexA*, or *Lz-LexA* was recombined with *13xLexAoP2-6XmCherry-HA*. Efficiencies of RNAi lines used in this study are indicated in Supplementary Table 1.

**Soda-lime, CO2, and O2 control experiments.** For the soda-lime treatment: the soda-lime (Sigma 72073) experiment was designed based on the previous study[48]. To avoid crowding, eight females and six male flies were crossed for all experiments and vials were shifted to new vials every day. Twenty soda-lime particles were wrapped and sealed with gauze (referred to as a soda-lime pocket). This soda-lime pocket was attached 5 mm above the food to diminish metabolic $CO_2$ emitted from larvae. To eliminate atmospheric $CO_2$, a 1000 μL pipette tip containing fifteen loosely-packed particles of soda-lime was inserted into a vial sealed with parafilm (Supplementary Fig. 1c). Putting more than twenty soda-lime particles in the pipette tip inhibits air flow and putting the pocket inside the food negatively affected larval growth. With this number of soda-lime particles, there was no developmental influence on larvae. Experiments were independently repeated at least three times.

For $O_2$ and $CO_2$ modulation experiments: hypoxia, hyperoxia and hypercapnia experiments were done in a hypoxia chamber (Modular Incubator Chamber MIC-101, Billups-Rothenberg.Inc or ProOX C21, BioSpherix). 10% (±0.5%) $O_2$ was used for hypoxia experiments, 40% (±0.5%) $O_2$ for hyperoxia, 13% (±0.5%) $CO_2$ for hypercapnia. *Drosophila* larvae were synchronized and cultured in normoxic conditions until 72 h after egg laying, and shifted to either hypoxic or hyperoxic condition. After rearing animals for 48 h in the chamber, wandering third-instar larvae were dissected immediately. For hypercapnia, the first-instar larvae were synchronized and shifted to the chamber, and dissected when they reached the wandering third-instar. Hypoxia/hyperoxia/hypercapnia and soda-lime experiments were done at 25 °C.

**Immunohistochemistry.** Lymph glands were dissected and stained as previously described[8]. Following primary antibodies were used in this study: αLz (DSHB, 1:10), αSima[49] (1:100), αβgal (Promega, 1:1000), αnc82 (DSHB, 1:10), αAntp (DSHB, 1:10), αp4EBP (Cell signaling, 2855 S, 1:100), αpAKT (Cell signaling, 4060 S, 1:100) and αSerrate (K. Irvine, 1:1000). Cy3-, FITC- or Alexa Fluor 647-conjugated secondary antibody (Jackson Laboratory) was used for staining. Alexa Flour 594 Phalloidin (Thermo Fisher, A12381) was used for F-actin staining. All samples were mounted in VectaShield (Vector Laboratory) and imaged by Zeiss Axiocam 503, Nikon C2 Si-plus or Zeiss LSM880 Airyscan confocal microscopy.

**Fig. 5** CO2SN/HypSNs systemically control Dilp6 and Serrate. **e–g** indicate the number of CCs in a single lymph gland lobe. Scale bar: 20μm; except in **a**, **b**: 50μm. n.s: not significant ($p > 0.01$). *$p < 0.01$; **$p < 0.001$; ***$p < 0.0001$. Error bars in **c–d** and **h–l**: standard deviation. Bars in graph **e–g**, **x**: the median. CO2SN inhibited conditions: −CO2R with shaded yellow in **f**, **j–l**. Soda-lime generated low $CO_2$ conditions: shaded blue in **d**, **g**. **a–d** Dilp6 secreted from the fat body is the second systemic signal. *dilp6* is expressed upon loss of CO2SN activity (*Gr63a-LexA, LexOp-Shi^{ts1}; Dilp6-gal4, UAS-GFP*) in the fat body (Dilp6, green) (**a**, **b**, **d**), but not the brain or blood (**c**). **e–l** *dilp6* is necessary and sufficient in extra CC formation. Overexpression of *dilp6* in the fat body is sufficient to induce CC differentiation (*ppl-gal4; UAS-dilp6*) (**e**). Loss of *dilp6* in the fat body reverts CCs in the CO2SN mutant background to wild-type numbers (*Gr63a-LexA, LexOp-Shi^{ts1}; ppl-gal4, UAS-dilp6^{RNAi}*) (**f**). This is also seen under soda-lime treatment conditions (*ppl-gal4; UAS-dilp6^{RNAi}*) (**g**). Expression of either *upd3* (**h**) or *sima* (**i**) in the brain is sufficient to enhance *dilp6* expression in the fat body (*Elav-gal4; UAS-upd3* or *Elav-gal4; UAS-sima*). Increased expression of *dilp6* is disrupted by: silencing HypSNs (*Gr63a-LexA, LexOp-Shi^{ts1}; Gyc89da-gal4, UAS-Shi^{ts1}*) (**j**), by *sima* RNAi in HypSNs (*Gr63a-LexA, LexOp-Shi^{ts1}; Gyc89da-gal4, UAS-sima^{RNAi}*) (**j**), upon inhibition of neuronal *upd3* (*Gr63a-LexA, LexOp-Shi^{ts1}; Elav-gal4, UAS-upd3^{RNAi}*) (**k**), or when *dome^{RNAi}* is expressed in the fat body (*Gr63a-LexA, LexOp-Shi^{ts1}; ppl-gal4, UAS-dome^{RNAi}*) (**l**). **m–x** Insulin receptor (InR) activation by Dilp6 increases Serrate expression. pAKT (pAKT, red) (**m–n**) and p4EBP (p4EBP, red) (**o–p**) are upregulated in the lymph gland upon CO2SN inhibition (*Gr63a-gal4; UAS-Gr21a^{RNAi}*). Compared with wild type (**q**), Serrate expression is substantially enhanced when CO2SN activity is lost (*Gr63a-gal4; UAS-hid,rpr*) (Serrate, red) (**r**). Neuronal expression of either *sima* (*Elav-gal4; UAS-sima*) (**s**) or *upd3* (*Elav-gal4; UAS-upd3*) (**t**), is sufficient to enhance Serrate protein expression. This phenotype is also seen when *dilp6* is overexpressed in the fat body (*ppl-gal4; UAS-dilp6*) (**u**). *Su(H)-LacZ* is activated in cells that receive active Notch signal. The number of such cells increases when CO2SN activity is attenuated (*Gr63a-gal4, Su(H)-LacZ; UAS-hid,rpr*) (*Su(H)-LacZ*, red) (**v–w**). Quantitation is shown in **x**

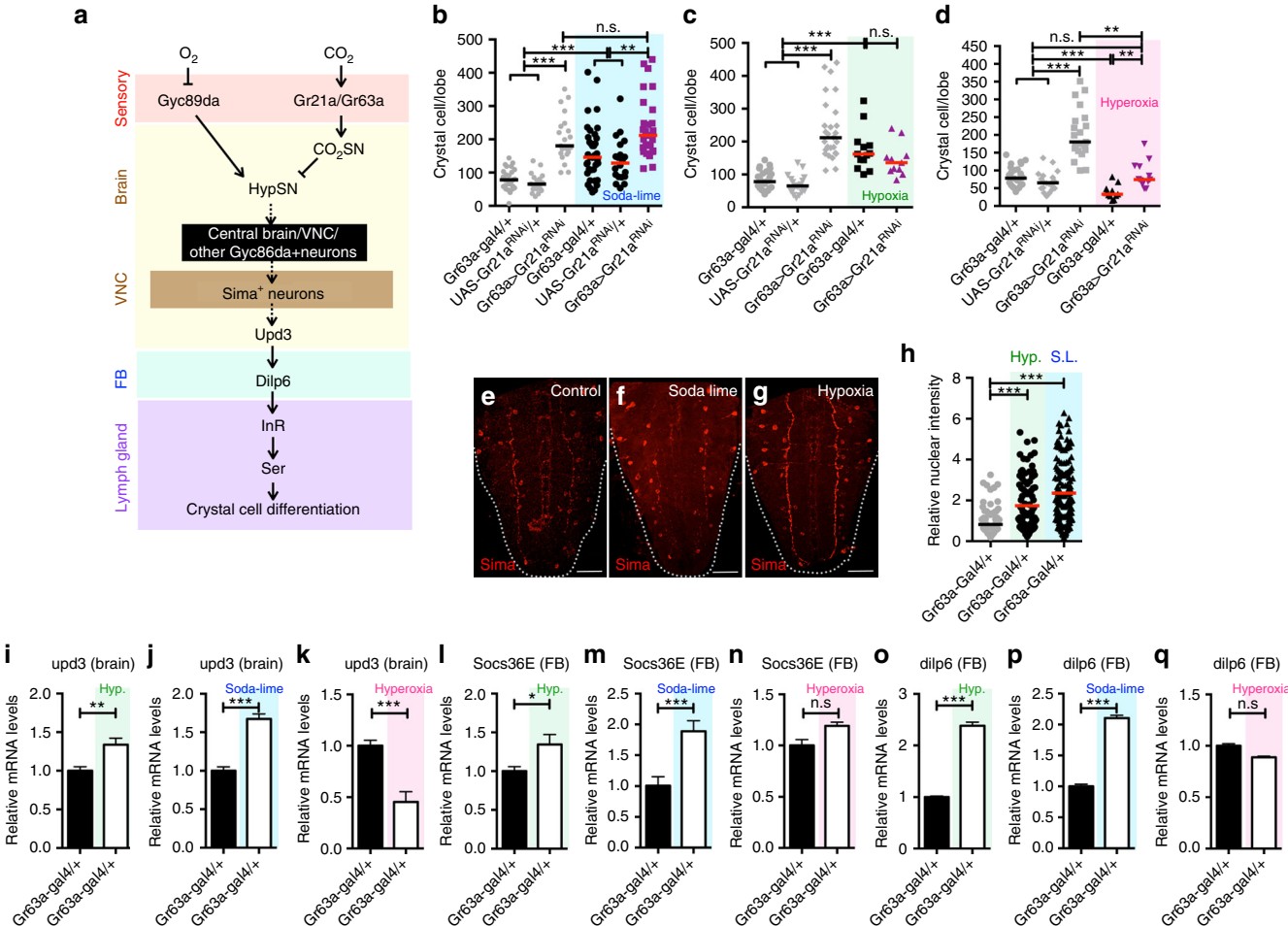

**Fig. 6** Environmental alterations support results of genetic manipulations. **b–d** indicate the number of CCs in a single lymph gland lobe. Scale bar: 20μm. n. s: not significant ($p > 0.01$). $*p < 0.01$; $**p < 0.001$; $***p < 0.0001$. Error bars in **i–q**: standard deviation. Bars in graph **b–d**, **h**: the median. Soda-lime generated low $CO_2$ conditions: blue; hypoxic (10% $O_2$) conditions: green; hyperoxic (40% $O_2$) conditions: pink. **a** A model summarizing the systemic response events. **b–d** Physiological relevance probed by altering environmental gases. Environmental $CO_2$ is scavenged by soda-lime. Hypoxic (<10% $O_2$) or hyperoxic (>40% $O_2$) conditions are achieved in hypoxia chambers. Elimination of $CO_2$ using soda-lime significantly increases the number of CCs in the lymph gland, and this response is further aggravated by the concurrent loss of $CO_2SN$ (*Gr63a-gal4; UAS-Gr21a^{RNAi}*) (**b**). Hypoxia increases CC number, but this phenotype cannot be further enhanced by low $CO_2SN$ activity (*Gr63a-gal4; UAS-Gr21a^{RNAi}*) (**c**). Hyperoxia on its own significantly suppresses CC development, and also efficiently rescues the low $CO_2SN$-induced CC differentiation to normal levels (**d**). **e–q** Phenotypic parallels between decreased levels of ambient $CO_2$ and $O_2$. In addition to low $CO_2SN$ activity induced by genetic manipulation (Fig. 3a–c), soda-lime treatment accumulates Sima protein expression in a specific subset of VNC neurons (**e**, **f**). Similar accumulation of Sima is observed by rearing larvae in hypoxic conditions (**g**). Quantitation shown in **h**. Both hypoxia and soda-lime treatments enhance *upd3* in the brain (**i-j**), *Socs36e* and *dilp6* transcript levels in the fat body (**l-m**, **o-p**). This expression is not observed in hyperoxia treatment (**k**, **n**, **q**)

For αSerrate staining, a pre-absorption step was essential for clear lymph gland staining. To do so, a 1:100 concentration (2% sodium azide) of antibody was incubated together with nine fixed larval cuticles overnight at 4 °C. Lymph glands were dissected at 72 h AEL and fixed in 3.7% formaldehyde for 25 min at room temperature. After fixation, lymph glands were washed 3 times (10 min each) nutating in 0.1% Tween20 in 1 × PBS and blocked in 1% BSA/0.1% Tween20 in 1xPBS for 30 min on a table-top shaker. Lymph glands were incubated overnight in αSerrate primary antibody (used at a final concentration of 1:1000) at 4 °C. Lymph glands were washed 3 times (10 min each) nutating in 0.1% Triton-X in 1xPBS and then incubated in Rat secondary antibody with 1% BSA/0.1% Triton-X in 1xPBS for 3 h at room temperature. After washing 3 times (10 min each) with 0.1% Triton-X in 1xPBS, samples were mounted in Vectashield with DAPI and imaged as described above.

**Quantitative real-time PCR analysis.** At least 20 larval organs (at least 100 for the lymph gland) were dissected to extract RNA. cDNA was synthesized with qPCR-RT kit (TOYOBO). qRT-PCR was performed by comparative $C_T$ method using SYBR Green Realtime PCR Master Mix (TOYOBO) and a StepOne-Plus Real-Time PCR detection thermal cycler (Applied Biosystems). Specific primers used for qRT-PCR are described in Supplementary Table 2.

**Quantification of samples.** CCs were quantified and analyzed by ImageJ (plug in: 3D object counter) or Imaris (Bitplane). CCs in individual primary lobes were counted for this study. Whole Z-stacks were compressed and analyzed for the quantification and figure presentation. Other stainings including pAKT, p4EBP or Serrate are shown in a single middle Z-stack slice. For CCs in circulation, *Lz-LexA LexAop-mCherry* positive blood cells in a larva were counted after bleeding. Prior to bleeding, animals were vortexed for 2 min to detach sessile population. Statistical significance of the CC phenotype was analyzed by Wilcoxon rank sum test after determining normality with the use of SPSS. Given natural variability of the number of CCs, we considered samples are significantly different only when $*p < 0.01$. Statistical results and genotypes are indicated in Supplementary Table 3.

Relative nuclear Sima intensity was analyzed with the use of IMARIS software. Amongst Sima immunoreactivity shown in the brain, we only selected high Sima+ cells to avoid background expressions, and of which nuclear intensity was measured and calculated. Relative intensity of mutants compared to wild type was presented in figures.

**Hypoxia tolerance experiment.** Three-day-old male flies were used for hypoxia tolerance experiments. 15 flies were placed in one empty vial and conditioned for 2 h before transferring to 1% oxygen-containing hypoxia chamber. Fly movement

was recorded for 1 h. 1 point was given when any fly from one vial shows a movement in 5 s; therefore, 12 point per 1 min for maximum. Flies were never placed in hypoxic condition before this experiment. Behavior assay was repeated more than three times with biologically independent samples.

**Data availability**. All data generated during and/or analyzed during the current study are included in this published article and its supplementary information files.

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

## Acknowledgements

The authors thank members of the Shim lab and the Banerjee lab for helpful discussions. The authors acknowledge the Bloomington, VDRC, DGRC, and KDRC *Drosophila* stock centers and the DSHB hybridoma bank. The authors thank the following individuals for stocks and reagents: C. Evans, H. Agaisse, A. Brand, E. Hafen, B. Lemaitre, T. Kitamoto, Y.N. Jan, G. Rubin, S. Artavanis-Tsakonas, D. Morton and K. Irvine. This work was supported by the Basic Science Research Program through the National Research Foundation (NRF) of Korea funded by the Ministry of Education (NRF-2014S1A2A2028388) and by the Ministry of Science, ICT and Future planning (NRF-2014R1A1A1002685) to J.S.; The UCLA part of this collaboration was supported by a Training Grant in Developmental Hematology (T32 HL086345) to C.M.S.; and the NHLBI grant R01 HL067395 and the Broad Stem Cell Research Center at UCLA to U.B.

## Author contributions

B.C., C.M.S., S.Y., N.C., and J.S. performed experiments; B.C., C.M.S., and J.S. analyzed data; B.C., C.M.S., U.B., and J.S. contributed to writing the manuscript; U.B. and J.S. conceived the idea and supervised the project.

## Additional information

**Competing interests:** The authors declare no competing interests.

