## [Peer Review File · Nature Communications]

REVIEWERS' COMMENTS:

Reviewer #1 (Remarks to the Author):

The authors have made a significant effort to address my previous comments and have addressed them satisfactorily. I would encourage including a table listing the various genotypes used, as well as the statistical results from the experiments, given how busy each graph is with multiple genotypes.

Reviewer #2 (Remarks to the Author):

While the article remains complex and the story a bit linear, the authors have adequately answer to nearly all my comments. I recommend its acceptance.

Reviewer #3 (Remarks to the Author):

I found this to be a solid and well analyzed manuscript. I had only minor comments:

"Our findings represent an ancestral link of environmental gas sensation and myeloid blood cell development." Is this conclusion really justified? If it isn't known how this works in vertebrates yet, how can it be clear that *Drosophila* shows an ancestral link? Right now, *Drosophila* provides an explanation. Once we know how things work in vertebrates, we can determine whether the two explanations are similar and then decide if how these systems evolved. You shouldn't jump to the conclusion of homology.

"As a hematopoietic readout, we count crystal cells (CCs), which function in wound healing, clotting, innate immunity and hypoxic stress response¹⁷⁻¹⁹." If the lymph glands aren't increasing in size, doesn't this suggest that it is only crystal cells that are being made, rather than hematopoiesis in general? This is just a CC readout.

"Finally, molting is associated with tracheal degeneration and represents an imbalance in the gaseous environment." It isn't really tracheal degeneration that happens – the inner lining of the tracheae is shed. Take a look at a cicada molt someday, as it is really amazing. The insect pulls the whole lining out of the tracheae and refills the tracheae with gas. Does this reduce oxygen tensions in the larva? In this section around line 229, it would be useful to have references that back up claims for changes in O₂ tension whenever described. If this is conjecture, then it would be useful to state that clearly.

"A wild-type number of crystal cells is generated through local developmental signals independent of sensory input^{23,29,30}." I wouldn't call this wild type because it can vary so much with environmental conditions. Perhaps you could just say that crystal cell number is regulated by the environment.

"These variations are tolerated during homeostasis, but further extremes need to be mitigated with a stress/innate immune response." I want to complain about the language here. To me, "tolerance" means the dose response of stress input to phenotype output. A tolerant animal would have a flat response curve. Is there any evidence, aside from this manuscript that the assertions are true? I don't think that there is any whole animal biology here explaining why the larva has this response. It is still molecular biology without biological context. The circulating levels of the crystal cells don't change; why will a reader care that the number of crystal cells changes in the lymph gland when crystal cells are presumably active in the circulation.

The authors show that transcripts for some antimicrobial peptides have baseline increases of 4-6x.

What does this mean for genes that can be induced 1000 fold when they are induced during an immune response. The increase may be statistically significant, but does it have any biological meaning?

I had a red-green colorblind postdoc once. He would have been unable to interpret most of the red on black pictures as he would have seen them as black on black. I'm wondering, if you have a monochromatic figure, why make it red on black instead of just grayscale?

REVIEWERS' COMMENTS:

Reviewer #1 (Remarks to the Author):

The authors have made a significant effort to address my previous comments and have addressed them satisfactorily. I would encourage including a table listing the various genotypes used, as well as the statistical results from the experiments, given how busy each graph is with multiple genotypes.

We listed all the complete genotypes and statistical results in Supplementary Table 3, and indicated in figure legends. We thank the reviewer for valuable comments and recommending acceptance.

Reviewer #2 (Remarks to the Author):

While the article remains complex and the story a bit linear, the authors have adequately answer to nearly all my comments. I recommend its acceptance.

Thanks for recommending acceptance.

Reviewer #3 (Remarks to the Author):

I found this to be a solid and well analyzed manuscript. I had only minor comments:

“Our findings represent an ancestral link of environmental gas sensation and myeloid blood cell development.” Is this conclusion really justified? If it isn't known how this works in vertebrates yet, how can it be clear that *Drosophila* shows an ancestral link? Right now, *Drosophila* provides an explanation. Once we know how things work in vertebrates, we can determine whether the two explanations are similar and then decide if how these systems evolved. You shouldn't jump to the conclusion of homology.

We have reworded to: “Our findings establish a link between environmental gas sensation and myeloid cell development in *Drosophila*. A similar relationship exists in humans, but the underlying mechanisms remain to be established.”

“As a hematopoietic readout, we count crystal cells (CCs), which function in wound healing, clotting, innate immunity and hypoxic stress response¹⁷⁻¹⁹.” If the lymph glands aren't increasing in size, doesn't this suggest that it is only crystal cells that are being made, rather than hematopoiesis in general? This is just a CC readout.

We show that the number of cells in the lymph gland (DAPI+) or plasmatocytes (Hml+) is not changed upon loss of CO₂ receptor (Supplementary figure 1m-s). The fact that the “size does not change” does not mean that the CCs (5% in normal) might not be formed at the cost of plasmatocytes (95% in normal).

“Finally, molting is associated with tracheal degeneration and represents an imbalance in the gaseous environment.” It isn't really tracheal degeneration that happens – the inner lining of the tracheae is shed. Take a look at a cicada molt someday, as it is really amazing. The insect pulls the whole lining out of the tracheae and refills the tracheae with gas. Does this reduce oxygen tensions in the larva? In this section around line 229, it would be useful to have references that back up claims for changes in O₂ tension whenever described. If this is conjecture, then it would be useful to state that clearly.

We changed the wording to:

“Finally, molting is associated with the shedding of the cuticular intima and degeneration of tracheoles that lack these inner linings. As the tracheal tube is filled with fluid and devoid of gases at this stage, we speculate that this process is also likely to alter oxygen tension in the hemolymph”

“A wild-type number of crystal cells is generated through local developmental signals independent of sensory input^{23,29,30}.” I wouldn’t call this wild type because it can vary so much with environmental conditions. Perhaps you could just say that crystal cell number is regulated by the environment.

We could say that, but that will be totally incorrect. There are local signals that determine an average number of crystal cells (about 5%). On top of local signals, change in environment alters the average in a consistent manner.

This is not so complicated, really. We do not all have the same number of T cells and the variation is large, so there is a normal (read wild type) range. During immune challenge, a certain repertoire increases to a much larger number. You do not conclude from this observation that T cell development is entirely dependent on immune challenge.

“These variations are tolerated during homeostasis, but further extremes need to be mitigated with a stress/innate immune response.” I want to complain about the language here. To me, “tolerance” means the dose response of stress input to phenotype output. A tolerant animal would have a flat response curve. Is there any evidence, aside from this manuscript that the assertions are true? I don’t think that there is any whole animal biology here explaining why the larva has this response. It is still molecular biology without biological context. The circulating levels of the crystal cells don’t change; why will a reader care that the number of crystal cells changes in the lymph gland when crystal cells are presumably active in the circulation.

We have removed this statement. Circulating blood is not a site of hematopoiesis and just as in the human bone marrow; immune challenges are met with increased activity at the site of hematopoiesis. These cells will be released into the early pupa and adult, and if the challenge is strong, into the larva as well.

The authors show that transcripts for some antimicrobial peptides have baseline increases of 4-6x. What does this mean for genes that can be induced 1000 fold when they are induced during an immune response. The increase may be statistically significant, but does it have any biological meaning?

The 1000 fold is a systemic response from the fat body in adult flies, not a cellular response from blood cells in either larvae or adults. We have added the phrase “the significance of this 4-6x response by the blood cells in the absence of any microbial infection is not clear”

I had a red-green colorblind postdoc once. He would have been unable to interpret most of the red on black pictures as he would have seen them as black on black. I’m wondering, if you have a monochromatic figure, why make it red on black instead of just grayscale?

This is a very good point that we will keep in mind for the future. We take this advice to heart and will try to achieve this next time. This manuscript is long delayed, so we wish not to hold it up for now.